# Facile generation of bridged medium-sized polycyclic systems by rhodium-catalysed intramolecular (3+2) dipolar cycloadditions

Bao-Long Hou[1,3], Jonathan J. Wong [2,3], Na Lv[1,3], Yong-Qiang Wang[1,3], K. N. Houk [2✉] &
Chuang-Chuang Li [1✉]

Bridged medium-sized bicyclo[m.n.2] ring systems are common in natural products and potent pharmaceuticals, and pose a great synthetic challenge. Chemistry for making bicyclo[m.n.2] ring systems remains underdeveloped. Currently, there are no general reactions available for the single-step synthesis of various bridged bicyclo[m.n.2] ring systems from acyclic precursors. Here, we report an unusual type II intramolecular (3+2) dipolar cycloaddition strategy for the syntheses of various bridged bicyclo[m.n.2] ring systems. This rhodium-catalysed cascade reaction provides a relatively general strategy for the direct and efficient regioselective and diastereoselective synthesis of highly functionalized and synthetically challenging bridged medium-sized polycyclic systems. Asymmetric total synthesis of nakafuran-8 was accomplished using this method as a key step. Quantum mechanical calculations demonstrate the mechanism of this transformation and the origins of its multiple selectivities. This reaction will inspire the design of the strategies to make complex bioactive molecules with bridged medium-sized polycyclic systems.

[1] Shenzhen Grubbs Institute, Department of Chemistry, Southern University of Science and Technology, Shenzhen, China. [2] Department of Chemistry and Biochemistry, University of California, Los Angeles, Los Angeles, CA, USA. [3]These authors contributed equally: Bao-Long Hou, Jonathan J. Wong, Na Lv, Yong-Qiang Wang. ✉email: houk@chem.ucla.edu; ccli@sustech.edu.cn

Molecules with bridged medium-sized ring systems[1,2] are advantageous for their pharmacological activity and for their selective and tight binding to biological targets. Bridged medium-sized bicyclo[m.n.1] and bicyclo[m.n.2] ring systems are strained and widely found in natural products with important biological activities (such as medicines Taxol, Picato, and Artemisinin; Fig. 1a)[3,4]. In contrast to the myriad approaches for the creation of bicyclo[m.n.1] ring systems, chemistry for bicyclo[m.n.2] ring systems remains underdeveloped. Bridged bicyclo[m.n.2] ring systems pose a great synthetic challenge and have prompted considerable interest from the synthetic community, leading to several remarkable syntheses[5–11]. However, so far there are no general reactions available for the single-step synthesis of various bridged bicyclo[m.n.2] ring systems from acyclic precursors. Thus, developing efficient reactions for achieving these bridged ring systems is very important.

Intramolecular cycloaddition reactions are very significant for efficiently creating polycyclic systems. The intramolecular Diels–Alder (IMDA) cycloaddition reaction is one of the most extensively used reactions for making ring systems[12–15]. IMDA cycloadditions are classified into type I and type II according to how the dienophile motif is linked to the diene. Type I IMDA cycloadditions (linked at the 1-position of the diene) are great for synthesizing fused bicyclo[m.4.0] ring systems. The pioneering Shea type II IMDA cycloadditions (linked at the 2-position of the diene) are powerful for the preparation of a few of bridged bicyclo[m.3.1] ring systems[15–17]; however, these are very rarely used for the creation of all-carbon bicyclo[m.2.2] ring systems[18], because the formation of such bicyclo[m.2.2] ring systems ($m = 3$, 4, or 5) also with an unfavorable strained bridgehead olefin[19] (Bredt's rule)[20] are usually more challenging than their regioisomeric products[21] (Fig. 1b). Particularly, there have been no reports of type II IMDA reactions being used to make bicyclo[5.2.2], bicyclo [4.2.2], and bicyclo[3.2.2] ring systems. In addition, other type II intramolecular cycloadditions[22,23] (including innovative Davies-[4+3][24], remarkable Wender-[4+4][25], and our [5+2][26,27]) are unknown for the synthesis of bicyclo[m.n.2] ring systems. Currently, few intramolecular cycloaddition reactions are available for the direct and efficient synthesis of various bicyclo[m.n.2] ring systems. An absence of direct procedures has impeded the in-depth evaluation of their potential pharmaceutical value. Therefore, it is still highly desirable to develop new and efficient strategies for constructing these attractive bridged bicyclo[m.n.2] ring systems.

The 1,3-diplor cycloaddition of transient carbonyl ylides generated from carbene is a very well-established reaction and it is also as an attractive strategy toward the synthesis of complex natural products[28–30]. Stimulated by the challenges of previous type II cycloadditions, we posited that an unusual rhodium-catalyzed type II intramolecular (3+2) dipolar cycloaddition might be achieved with different types of substrates to synthesize bicyclo[m.n.2] ring systems without strained bridgehead olefins (Fig. 1c). Specifically, we considered an N-sulfonyl-1,2,3-triazole moiety and an unactivated alkene both tethered at the α-position of the carbonyl group of an aldehyde with various tether lengths (A, Fig. 1c). Treatment of N-sulfonyl-1,2,3-triazole A with rhodium catalysts would generate the rhodium-iminocarbene B[31,32]. The iminocarbene B would give reactive 1,3-dipole C. The 1,3-dipole and alkene group in C would undergo the desired (3+2) dipolar cycloaddition via intermediate D, to provide various useful and highly functionalized bridged bicyclo[m.n.2] ring systems (e.g., bicyclo[5.2.2], bicyclo[4.2.2], bicyclo[3.2.2], and azabicyclo[3.2.2] ring systems). The anticipated cycloaddition described herein is different from the previous examples because it does not produce anti-Bredt double bond. This is the reason why smaller rings could be accessed compared to the other type II

cycloadditions. Remarkably, these bridged skeletons contain medium-sized ring systems[1,2], which have high strain energy. It is very difficult to form such ring systems and doing so depends on the reactivities of the corresponding acyclic precursors and their steric effects because of unfavorable transannular interactions and enthalpic and entropic effects[33,34]. The carbenes derived from the corresponding N-sulfonyltriazoles have been used in cycloaddition reactions to synthesize fused all-carbon ring systems[35–37]. So far, there have been no reports of intramolecular dipolar cycloaddition reactions for the synthesis of all-carbon bicyclo[m.n.2] ring systems[38]. Particularly, the activation free energy of type II (3+2) cycloadditions is higher than that of type I cycloadditions, because of the strain inherent to the formation of the bridged ring systems[15,22]. Furthermore, alkene groups could potentially undergo competitive intramolecular cyclopropanation[39] with the reactive rhodium-carbene in intermediate B to give unexpected products. Meanwhile, the rhodium-carbene in B can also undergo intramolecular insertion into the C-H bond of alkanes[40,41]. In addition, the alkene group in C can competitively undergo intramolecular (5+2) cycloaddition involving the carbonyl ylide and the N-tosylimine. Similar to type II IMDA cycloadditions, the 1,3-dipole and alkene group in C can also undergo regioisomeric (3+2) cycloaddition to give bicyclo[m.n.1] ring system through intermediate E. All these competitive reactions make the desired type II (3+2) cycloaddition reaction more challenging. Furthermore, it was uncertain as to which diastereomer would form and if the cycloaddition would follow a concerted or stepwise pathway. Herein we report rhodium-catalyzed type II intramolecular (3+2) cycloadditions of N-sulfonyl-1,2,3-triazoles. These unique reactions provide a new strategy for the direct and efficient synthesis of various functionalized bridged medium-sized polycyclic systems and enable asymmetric total synthesis of nakafuran-8.

## Results and discussion

**Reaction development.** To test our hypothesis, initially, the 1,2,3-triazole **1a** (Fig. 2) was synthesized as the model substrate to identify feasible reaction conditions [see Supporting Information (SI) for details]. Treatment of **1a** with $Rh_2(Oct)_4$ (5 mol%) in 1,2-dichloroethane (DCE) at 85 °C for 3 h yielded the corresponding imine product **2a**. Compound **2a** was unstable to chromatography on both silica gel and aluminum oxide and was directly reduced by $LiAlH_4$ in a one-pot sequence to give the desired bicyclo[4.2.2] ring system **3a** as a single diastereomer in 81% isolated yield. The structure of **3a** was confirmed by X-ray diffraction. Notably, this cycloaddition constructed three new chemical bonds (highlighted in blue in **2**), two carbocyclic rings (highlighted in red in **3**) and one heterocyclic ring, and three new stereogenic centers, including one bridgehead all-carbon quaternary stereogenic center in a single step. This result is particularly significant, because installation of a bridgehead quaternary stereogenic center within a bridged bicycle is very challenging and is highly valuable in synthesis[42,43]. This reaction affords the opportunity to efficiently construct these bridgehead all-carbon quaternary stereogenic centers in natural products, such as nakafuran-8, PF-1018, tubingensin B, and calyciphylline C (C* in Fig. 1a).

**Substrate scope.** With the optimized conditions in hand, the generality of this strategy was examined with various substrates, as summarized in Fig. 2. First, we varied the $R^1$ groups and $R^2$ substituents on the internal position of the alkenes, affording bicyclo[4.2.2] ring systems **3b–r** in moderate-to-excellent yields (61–90%). The reactions proceeded well for aromatic substituents on the internal positions of the alkenes with electron-withdrawing or electron-donating groups in para or meta position of aryl groups, forming **3b–h** with 67–80% yields. Pleasingly, the methyl-substituted

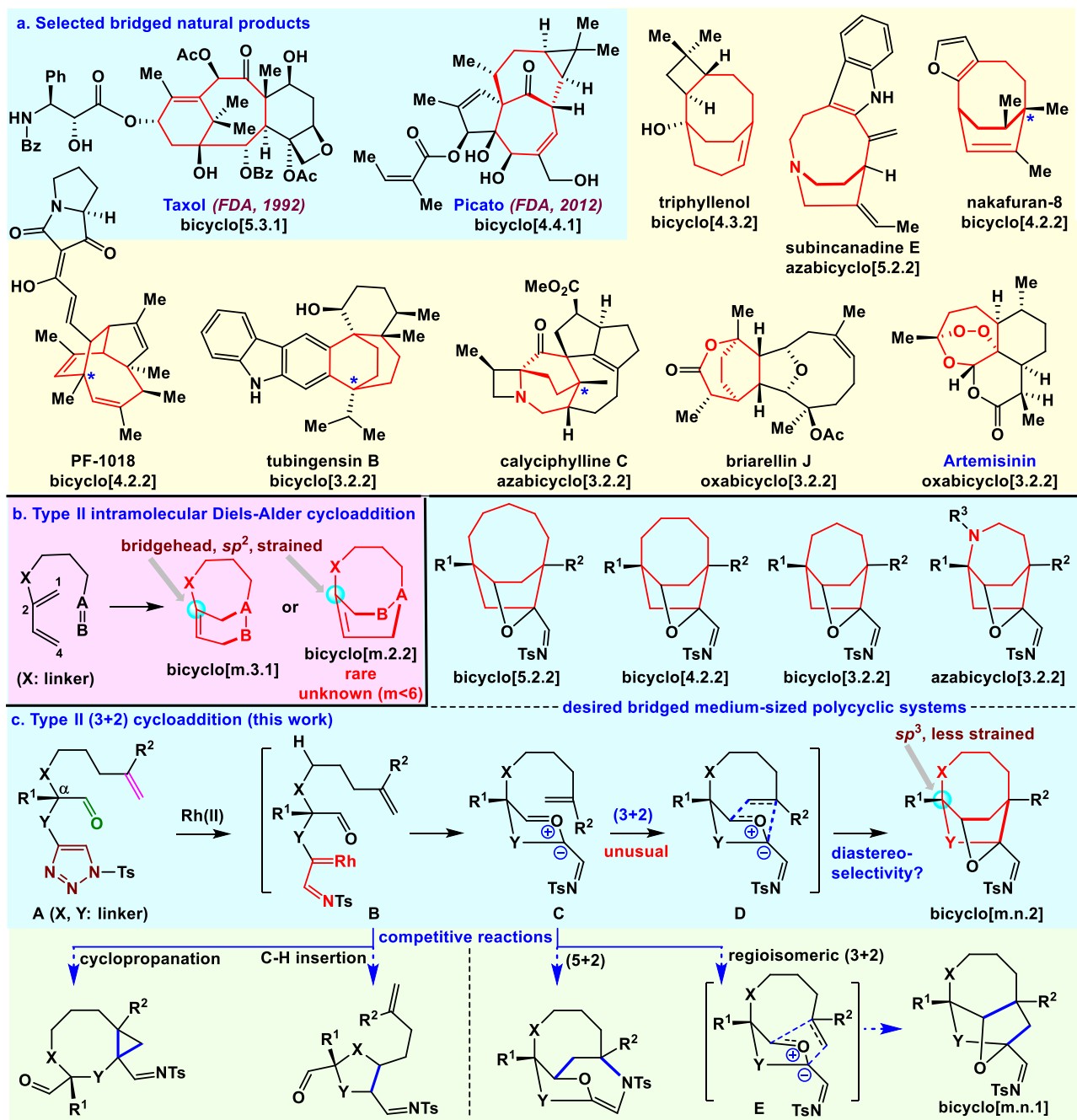

**Fig. 1 Bridged medium-sized ring systems and challenges of type II cycloadditions. a** Bridged medium-sized ring systems are highly valued in many compounds, including in some natural products that also serve as medicines. **b** Type II IMDA cycloaddition is used to make a few of bicyclo[m.3.1] ring systems but rarely used for the syntheses of bicyclo[m.2.2] ring systems, and all these bridged ring systems have a highly strained bridgehead olefin. **c** Type II (3+2) cycloaddition reaction (this work), their use in creating bicyclo[m.n.2] ring systems, and their potential competitive reactions. Direct generation of bridged ring systems by an intramolecular cycloaddition is called type II cycloaddition. Bridged ring systems are highlighted in red. Bz benzoxyl, Ac acetyl, Ts tosyl.

substrate **1i** delivered the desired product **3i** in 61% yield. Notably, the bridged bicyclo[4.2.2] ring system **3o** ($R^1 = H$, $R^2 = Me$) was obtained in an acceptable 61% yield. This finding shows that this cycloaddition probably does not rely on the Thorpe–Ingold effect, which is a considerable advantage. Remarkably, substrates **1s–z** with a fused four- or five-membered ring, all led to formation of **3s–z** with diverse stereochemistry in good yields, respectively. Particularly, the mono-substituted alkenes **1x** and **1z** yielded **3x** and **3z**, respectively. It is worth mentioning that the functionalized bridged ring system **3z** is a potential advanced intermediate for the total synthesis of

triphyllenol (Fig. 1a). Pleasingly, heterocycle-substituted products **3p–r**, which are good handles for further functionalization, were prepared in 75–90% yields. Rewardingly, several protecting groups, including triisopropylsilyl (TIPS), pivaloyl, benzyl, and trimethylsilyl groups, were tolerated very well. Furthermore, the pivaloyl group was potentially removed under $LiAlH_4$ conditions. Consequently, the imine products **2j–l** were hydrolyzed with wet basic alumina to give the corresponding aldehydes **3j–l**, respectively, in good yields, offering potential for further functional group transformations.

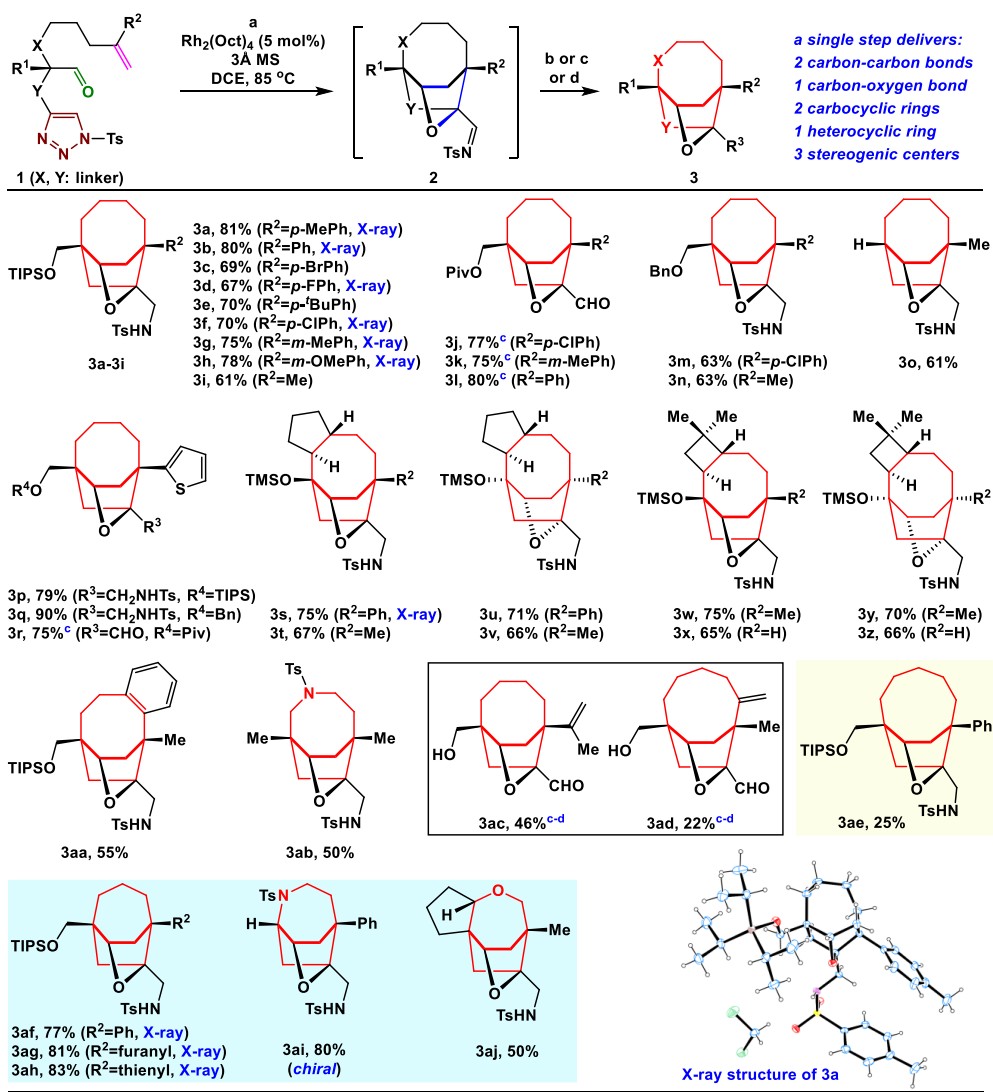

**Fig. 2 Substrate scope.** Rhodium-catalyzed type II (3+2) cycloaddition reaction using various substrates to give highly functionalized bridged medium-sized ring systems. Bn benzyl, Piv pivaloyl, TMS trimethylsilyl.

Notably, two substrates containing a phenyl ring or a nitrogen as the tether were also compatible (Fig. 2), and the desired phenyl-fused bicyclo[4.2.2] **3aa** or azabicyclo[4.2.2] **3ab** were obtained. Interestingly, treatment of **1ac** ($R^1 = CH_2OTIPS$, $R^2 =$ isopropenyl) with Rh$_2$(Oct)$_4$ (5 mol%) at 85 °C, followed by a wet basic alumina and tetrabutylammonium fluoride gave the desired bicyclo[4.2.2] ring system **3ac** and the unexpected bicyclo[5.2.2] ring system **3ad** in yields of 46 and 22%, respectively. Compound **3ac** with a bridgehead isopropenyl group has good potential for further modification. Furthermore, the highly strained bicyclo[5.2.2] ring system **3ae** with a difficult-to-form medium-sized 9-membered ring was also obtained, albeit with a 25% yield. The successful formation of bicyclo[5.2.2] ring systems **3ad** and **3ae** will provide useful ideas for designing the synthetic route of subincanadine E (Fig. 1a). It is noteworthy that the type II IMDA reaction conditions for unactivated dienophiles were usually very harsh (400–500 °C, gas phase)[15–17]; our type II (3+2) cycloaddition conditions for unactivated alkene dipolarophiles are much milder, which will be a significant advantage for their further application.

Encouraged by the successful syntheses of bicyclo[4.2.2] ring systems, we set out to investigate the feasibility of using the current strategy to make other bicyclo[m.2.2] skeletons with various tether lengths. As shown in Fig. 2, the corresponding precursors **1af–aj** with different tether lengths could undergo the rhodium-catalyzed type II (3+2) cycloaddition reaction, thus delivering a series of unique, functionalized bicyclo[3.2.2] skeletons **3af–aj**. Alkenes with various $R^2$ groups, including phenyl, furanyl, thienyl, and methyl groups, could take part in the cycloaddition. The desired bridged bicyclo[3.2.2] ring systems **3af–ah** with different functional groups were obtained in 77–83% yields. To our delight, the chiral azabicyclo[3.2.2] product **3ai** and the highly functionalized oxabicyclo[3.2.2] ring system **3aj** were obtained in good and acceptable yields, respectively. These results are particularly important, because the synthetically challenging azabicyclo[3.2.2] and oxabicyclo[3.2.2] ring systems are also found in some natural products, such as calyciphylline C and briarellin J, respectively (Fig. 1a). The structures of **3b**, **3d**, **3f–h**, **3s**, and **3af–ah** and the derivatives of **3p**, **3t**, **3w**, and **3x** were unambiguously confirmed by X-ray diffraction (see SI for details). To our knowledge, this work represents the first example of an intramolecular (3+2) dipolar cycloaddition for constructing bicyclo[3.2.2], bicyclo [4.2.2], and bicyclo[5.2.2] ring systems.

**Fig. 3 Asymmetric total synthesis of nakafuran-8.** DIPEA N,N-diisopropylethylamine, DMP Dess–Martin periodinane, LDA lithium diisopropylamide, DIBAL diisobutylaluminium hydride, KHMDS potassium bis(trimethylsilyl)amide, LiHMDS lithium bis(trimethylsilyl)amide.

**Asymmetric total synthesis of nakafuran-8.** After establishing the type II (3+2) cycloaddition reaction to make various bridged medium-sized ring systems, we moved toward the asymmetric total synthesis of nakafuran-8 (Fig. 3). Nakafuran-8 is an anti-feedant against common reef fishes[44] and has a remarkable inhibitory activity against human protein tyrosine phosphatase 1B, an enzyme involved in insulin signal regulation[45]. Structurally, nakafuran-8 has a unique bicyclo[4.2.2]decadiene skeleton with a furan ring and a bridgehead all-carbon quaternary stereocenter, which presents a formidable synthetic challenge. In addition, its absolute configuration has not been determined. Construction of the desired bicyclo[4.2.2] ring system showed very poor regioselectivity in the previous synthesis[5,6]. To date, asymmetric synthesis of nakafuran-8 has not been achieved.

The readily available chiral compound 4[46] underwent a diastereoselective aldol reaction with aldehyde 5[47] followed by TIPS protection of the resulting hydroxyl group to give compound 6. Reduction of 6 with NaBH$_4$ and subsequent oxidation provided 7 in 55% overall yield (20 g scale). The alkyne group in 7 reacted with TsN$_3$ to give 8 in 85% yield (10 g scale). The rhodium-catalyzed type II (3+2) cycloaddition of 8 with a N-sulfonyl-1,2,3-triazole moiety, an unactivated alkene and an aldehyde, provided the desired product 9, which underwent hydrolysis and reduction in a one-pot sequence to give alcohol 10 in 50% overall yield (gram scale). Compound 10 was converted to the corresponding iodide followed by cleavage of the C-O bond to generate alkene 11 in 63% overall yield (gram scale). Subsequent oxidation of 11 with Dess–Martin periodinane (DMP) gave the expected ketone. Diastereoselective and regioselective methylation at the α-position of the resultant ketone group with lithium

diisopropylamide and MeI and subsequent reduction provided 12 in 60% overall yield over 3 steps (gram scale). The absolute configurations of 11 and 12 were confirmed by the X-ray diffraction analysis of their derivatives (see SI). Accordingly, compound 12 underwent standard Barton deoxygenation smoothly and then TIPS deprotection to give 13 in 62% overall yield (gram scale). DMP oxidation of 13 followed by the formation of furanone and chemoselective olefin isomerization developed by Shenvi[48] produced compound 14. Treatment of 14 with DIBAL gave nakafuran-8 in 65% yield, completing asymmetric total synthesis of nakafuran-8. Thus, the absolute configuration of nakafuran-8 was unambiguously established by our total synthesis.

**Quantum mechanical study of mechanisms and selectivities.** We carried out a computational exploration of the regioselectivity, chemoselectivity, and diastereoselectivity of the (3+2) cycloaddition between an alkene dipolarophile and a carbonyl ylide 1,3-dipole. Quantum mechanical calculations (Fig. 4) were carried out with density functional theory and solvation energies computed at the ωB97X-D/6-311 + G(d,p),CPCM(DCE) level of theory. Conformational analysis was carried out with CREST. All (3+2) cycloadditions were found to proceed through concerted but highly asynchronous mechanisms.

The three-dimensional structures of the lowest energy transition states (TSs) are shown in Fig. 4a. Both involve typical intramolecular carbonyl ylide cycloadditions to produce functionalized bridged ring systems. The favored cycloaddition leads to a bicyclo[4.2.2] ring system (highlighted in red) in 15. In this reaction, the most nucleophilic unsubstituted terminus of the

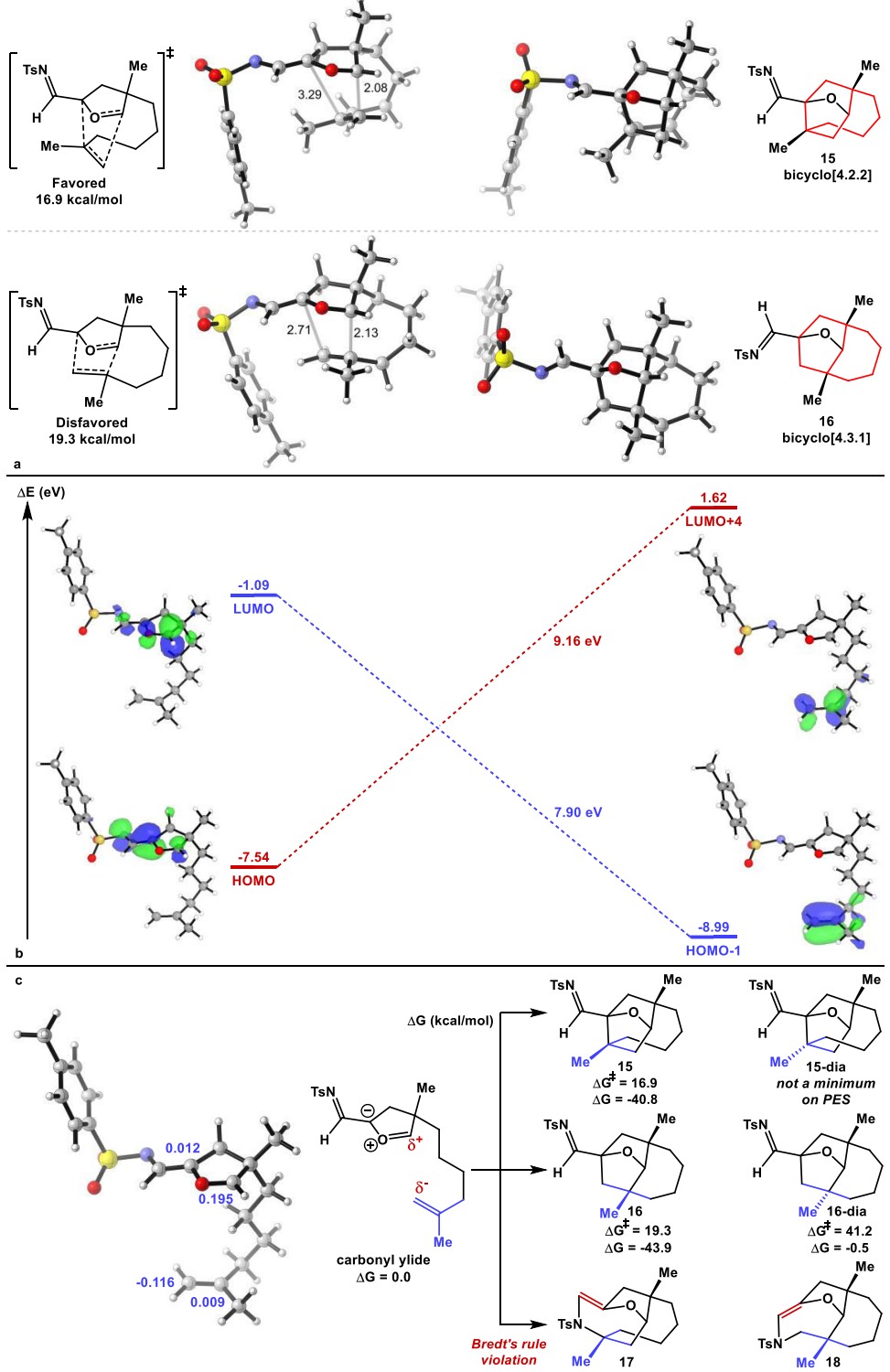

**Fig. 4 Quantum mechanical calculations of the carbonyl ylide (3+2) cycloaddition. a** Optimized structures of the two favored transition states. **b** FMO interactions contribute to the stabilization of the TS for the major product. **c** Energetics of (3+2) and (5+2) cycloadditions and charge distribution of the carbonyl ylide.

alkene attacks the most electrophilic terminus of the carbonyl ylide. The disfavored cycloaddition leads to the bicyclo[4.3.1] ring system (highlighted in red) in **16**, with a mismatch of electrophilic and nucleophilic sites in the TS. Both have the connecting linker *anti* to the O of the carbonyl ylide (*exo*). Although the favored TS is highly asynchronous, no intermediate

could be located. As expected, the disfavored regioisomeric TS is less asynchronous because of more balanced, but overall weaker, frontier molecular orbital (FMO) interactions[49–52].

The FMOs of the carbonyl ylide and the alkene dipolarophile are shown in Fig. 4b. These calculations show that the strongest interaction is between the π bonding orbital on the olefin (highest

occupied molecular orbital (HOMO) − 1) and the π* on the carbonyl ylide (lowest unoccupied molecular orbital (LUMO)), which have an energy difference of 7.90 eV. The allyl anion-like (HOMO) of the carbonyl ylide and the π* orbital on the olefin (LUMO + 4) interact less strongly with an energy difference of 9.16 eV between the two orbitals. As expected, the HOMO of the carbonyl ylide resembles a normal allyl anion-like HOMO, with a node at O and highly polarized toward the terminus bearing the electron-withdrawing substituent. The LUMO is strongly polarized in the opposite direction. The major FMO interaction is between the carbonyl ylide LUMO and the alkene HOMO, which has the largest coefficient at the unsubstituted terminus, consequently the most nucleophilic site. Shown in blue in Fig. 4c are the Hirshfeld charges corresponding to each of these atoms. Most evident is the significant negative charge on the terminal methylene of the olefin (−0.12) and the positive charge on the carbonyl ylide carbon (0.20). The electrostatic interaction between these two carbons in the TS further influences the regioselectivity of this reaction.

Reactants, TSs, and products were optimized for the four possible (3+2) cycloadditions and for two of the forbidden (5+2) cycloadditions. These results are also shown in Fig. 4c. The major *exo*-product **15** discussed above and its corresponding TS barrier were calculated, revealing an extremely exothermic transformation with a barrier of only 16.9 kcal/mol and overall Gibbs free energy change of −40.8 kcal/mol. The diastereomer **15-dia** in which the two methyl groups are *trans* to one another, resulting from the opposite facial selectivity, is geometrically impossible and was not found to be a minimum on the potential energy surface. The activation barrier and the overall free energy change of the regioisomeric *exo*-product **16** were computed to be 19.3 kcal/mol and −43.9 kcal/mol, respectively. This 2.4 kcal/mol higher activation barrier means that only a few percent of this adduct should be formed. The alternate diastereomer **16-dia** with *trans* methyl groups is geometrically highly contorted but could be located with an energy barrier of 41.2 kcal/mol and an overall exothermicity of only −0.5 kcal/mol. Finally, the products **17** and **18** of the (5+2) reaction involving the carbonyl ylide and the *N*-tosylimine were calculated to be relatively unstable with overall Gibbs free energy changes of −14.3 kcal/mol and −26.2 kcal/mol for the two regioisomers. We did not attempt to find TSs for these highly disfavored reactions.

**Summary**. In summary, we have developed a rhodium-catalyzed type II (3+2) intramolecular cycloaddition cascade reaction of a *N*-sulfonyl-1,2,3-triazole, involving a carbonyl group and a simple unactivated alkene. This unusual reaction will be an approach complementary to type II IMDA cycloaddition to access bridged polycyclic systems. The reaction proceeds chemoselectively, regioselectively, and diastereoselectively, thus establishing a new and relatively general strategy for the efficient and straightforward synthesis of various functionalized and synthetically challenging bridged medium-sized ring systems (e.g., bicyclo[5.2.2], bicyclo[4.2.2], azabicyclo[4.2.2], bicyclo[3.2.2], azabicyclo[3.2.2], and oxabicyclo[3.2.2] ring systems). Particularly, this work represents, to our knowledge, the first example of a rhodium-catalyzed intramolecular (3+2) cycloaddition to make synthetically challenging eight-membered ring systems[53] and represents the first example of an intramolecular dipolar cycloaddition for constructing all-carbon bicyclo[m.2.2] ring systems. To the best of our knowledge, the first asymmetric total synthesis of nakafuran-8 was achieved using this method. Quantum mechanical calculations show the nature of the TSs for these reactions and the origins of regioselectivity and stereoselectivity. We believe that the reaction developed here will inspire the design of new strategies to make complex natural products and other bioactive molecules with bridged medium-sized polycyclic systems.

## Methods

**General experimental procedure**. An oven-dried tube was charged with triazole compound **1** (0.1 mmol, 1.0 equiv.), 3 Å MS (30 mg), and $Rh_2(OCt)_4$ (4 mg, 0.005 mmol, 0.05 equiv.). The tube was evacuated and backfilled with argon (repeated three times). Then DCE (2 mL, 0.05 M/L) was added into the reaction via syringe. The reaction mixture was stirred at 85 °C for 3 h. The solution was then cooled to 25 °C. Tetrahydrofuran (THF; 4 mL) was added into the reaction via syringe. The solution was cooled to 0 °C. $LiAlH_4$ (1.0 M in THF, 0.2 mL, 0.2 mmol, 2.0 equiv.) was added dropwise and the solution was stirred at 0 °C for 1.5 h. Following quenching with saturated aqueous potassium sodium tartrate (6 mL), the reaction mixture was extracted by EtOAc (30 mL × 3), the combined organic layers were washed with saturated brine (10 mL), dried over $Na_2SO_4$, concentrated in vacuum, and purified by column chromatography on silica gel to give the products.

## Data availability

The data supporting the findings of this study are available in the paper and its Supplementary Information; further data are available from the corresponding author on request. The X-ray crystallographic coordinates for structures reported in this study have been deposited at the Cambridge Crystallographic Data Centre (CCDC), under deposition numbers CCDC 2065 575 (**3a**), CCDC 2068651 (**3b**), CCDC 2053925 (**3d**), CCDC 2053921 (**3f**), CCDC 2053927 (**3g**), CCDC 2053929 (**3h**), CCDC 2053972 (**3p'**), CCDC 2054598 (**3s**), CCDC 2054000 (**3t'**), CCDC 1920650 (**3w'**), CCDC 1920646 (**3x'**), CCDC 2053961 (**3af**), CCDC 2053962 (**3ag**), CCDC 2053964 (**3ah**), CCDC 2054002 (**11'**), CCDC 2054003 (**12'**). These data can be obtained free of charge from the Cambridge Crystallographic Data Centre via www.ccdc.cam.ac.uk/data_request/cif.

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

## Acknowledgements

This work was supported by the National Natural Science Foundation of China (Grant no. 21971105), the Shenzhen Nobel Prize Scientists Laboratory Project (C17783101), Science and Technology Key Project of Guangdong Province (2020B1111110004), and Research Projects of Universities of Guangdong Province (2019KZDXM005). K.N.H. is also grateful to the National Science Foundation (CHF-1764328) for financial support. Calculations were performed on the Hoffman2 cluster at the University of California, Los Angeles, and the Extreme Science and Engineering Discovery Environment (XSEDE), which is supported by the National Science Foundation (Grant OCI-1053575).

## Author contributions

C.-C.L. conceived of and directed the project. B.-L.H., N.L. and Y.-Q.W. performed the synthetic experiments, developed the reactions, and finished the asymmetric total synthesis of nakafuran-8. J.J.W. performed the computational experiments and provided mechanism analysis. K.N.H. directed the computational calculations and mechanism analysis. All authors analyzed and discussed the results. K.N.H. and C.-C.L. prepared the manuscript.

## Competing interests

The authors declare no competing interests.
