## [Peer Review File · Nature Communications]

Reviewers' Comments:

Reviewer #1:

Remarks to the Author:

The manuscript reports a novel family of reactions for the rhodium-catalyzed synthesis of bicyclic systems. The synthetic work is complemented by a computational study on the mechanism and the origin of selectivity. The topic is sufficiently relevant for Nat Comm and the work is technically correct.

I have only two minor comments on the computational part:

1. I am mildly curious about the selection of functional for the DFT calculations. If I am not mistaken, Houk's group has been favoring M06-2X throughout the years, but here they shift to wB97X-D. Is there any reason for this choice?
2. The Supporting Information indicates that conformational analysis was carried out with CREST. This is omitted from the main text, but I think it is a useful information that further strengthens the reliability of the work

Reviewer #2:

Remarks to the Author:

This is an interesting paper worthy of publication in Nature Communications. Dr. Chuang-Chuang is a world leader in the utilization of type-II intramolecular cycloadditions in total synthesis. His traditional system has been 5+2 cycloadditions. However, here he has recognized that carbonyl ylides derived from the intramolecular reactions of a carbenes with a carbonyl, could be arranged to undergo a follow up type-B 3+2 cycloaddition, leading to the rapid synthesis of bridged tricyclic products. The scheme works very well and its application was demonstrated in a total synthesis of a natural product (it required 11 additional steps so there was a need of a lot of functional group manipulations but still was a nice demonstration. A collaboration with the Houk group enabled a computational rational for the stereochemical outcome to be obtained. Generally, I am in favor of this manuscript, but there are a few things that need to be changed, especially in the introduction.

The introduction needs to have a proper balance of the literature background:

1. The 1,3-dipolar cycloaddition of transient carbonyl ylides generated from carbene is a very well-established reaction. It needs to be properly introduced and not just mentioned in passing. A Padwa review was referenced but there to be at least a few sentences and acknowledging the scope and application in synthesis, and confirming that it has never been used in type II cycloaddition (if that is indeed the case).
2. The cycloaddition described herein is different from the previous examples because it does not produce an anti-Bredt double bond. This is the reason why smaller rings can be accessed compared to the other type II cycloadditions and this should be stated
3. There should be a mention of the background about the uses of carbenes derived from N-sulfonyltriazoles in cycloaddition reactions.
4. There is a big analysis of all the possible carbene derived products that could have been formed. However, most of these products were not especially likely so the discussion is a bit of an exaggeration. The formation of the carbonyl ylide was highly likely and if the alkene was suitably placed the cycloaddition would be expected to occur. Li's recognition of the opportunity was clever but the outcome is what one would have expected.

Finally, the draft in its current form has a lot of grammatical errors and clumsy sentences. The schemes look great but someone needs to go through the text very carefully.

Reviewer #3:

Remarks to the Author:

The manuscript authored by Houk, and Li et al. describes a new synthetic approach to bridged

polycyclic compounds based on the application of catalytic intramolecular [3+2] cycloadditions. A wide range of functionalized [m.n.2] bridged scaffolds can be accessed using this method. The utility of this practical methodology was fully demonstrated by completion of the 17-step asymmetric total synthesis of nakafuran-8 highlighted by the rapid assembly of the challenge all carbon [4.2.2] bridged ring system. Besides, detailed mechanistic study of the regioselectivity, chemoselectivity, and diastereoselectivity of the [3+2] cycloaddition was provided through DFT calculations. Since synthesis of bridged medium-sized bicyclo[m.n.2] rings via Type II intramolecular [3+2] cycloadditions has been even rarer, this work is of great importance in the field of cycloaddition reactions and therefore the reviewer would be happy to support its publication in Nature Communications. This work shows a complete story from method development, mechanistic study and synthetic applications. I believe it can be published as is.

Answers to reviewers

We have revised our manuscript based on the reviewers' comments and the detail is listed below. Please see the Related Manuscript file of “Text-revised-marked by the authors” for the highlighted changes.

Reviewers 1:

Recommendation: The topic is sufficiently relevant for Nat Comm and the work is technically correct.

Comments: The manuscript reports a novel family of reactions for the rhodium-catalyzed synthesis of bicyclic systems. The synthetic work is complemented by a computational study on the mechanism and the origin of selectivity. The topic is sufficiently relevant for Nat Comm and the work is technically correct.

Answer: We highly appreciate this reviewer's comments.

Question 1: I am mildly curious about the selection of functional for the DFT calculations. If I am not mistaken, Houk's group has been favoring M06-2X throughout the years, but here they shift to wB97X-D. Is there any reason for this choice?

Answer: Yes, you are right, the Houk group has been gravitating toward wB97X-D lately, on the way to wB97M-V, in part because of the extensive benchmarks by Mardirossian and Head-Gordon (Mardirossian, N. & Head-Gordon, M. Thirty years of density functional theory in computational chemistry: an overview and extensive assessment of 200 density functionals. *Molecular Physics*. **115**, 2315-2372 (2017)). This indicates that, in many cases, wB97X-D is more accurate than M06-2X when calculating activation barriers of a wide variety of systems. For this work, the activation barriers were calculated with M06-2X too, to verify consistency across functionals.

From these calculations, it is apparent that, although the absolute energy values are different, both results are consistent with experimental observations.

Question 2: The Supporting Information indicates that conformational analysis was carried out with CREST. This is omitted from the main text, but I think it is a useful information that further strengthens the reliability of the work

Answer: We would like to thank the reviewer for his or her helpful suggestion and we have added this sentence to Page 12 in the main text of the manuscript.

Reviewers 2:

Recommendation: This is an interesting paper worthy of publication in Nature Communications.

Comments:

This is an interesting paper worthy of publication in Nature Communications. Dr. Chuang-Chuang is a world leader in the utilization of type-II intramolecular cycloadditions in total synthesis. His traditional system has been 5+2 cycloadditions. However, here he has recognized that carbonyl ylides derived from the intramolecular reactions of a carbenes with a carbonyl, could be arranged to undergo a follow up type-B 3+2 cycloaddition, leading to the rapid synthesis of bridged tricyclic products. The scheme works very well and its application was demonstrated in a total synthesis of a natural product (it required 11 additional steps so there was a need of a lot of functional group manipulations but still was a nice demonstration. A collaboration with the Houk group enabled a computational rational for the stereochemical outcome to be obtained. Generally, I am in favor of this manuscript, but there are a few things that need to be changed, especially in the introduction. The introduction needs to have a proper balance of the literature background:

Answer: We highly appreciate this reviewer's comments.

Question 1: The 1,3-dipolar cycloaddition of transient carbonyl ylides generated from carbene is a very well-established reaction. It needs to be properly introduced and not just mentioned in passing. A Padwa review was referenced but there to be at least a few sentences and acknowledging the scope and application in synthesis, and confirming that it has never been used in type II cycloaddition (if that is indeed the case).

Answer: We highly appreciate this reviewer's comments. Regarding the issues addressed by this reviewer, we have provided some discussion about the 1,3-dipolar cycloaddition of transient carbonyl ylides generated from carbene. A discussion has been added in the text in Page 4, as the following:

“The 1,3-dipolar cycloaddition of transient carbonyl ylides generated from carbene is a very well-established reaction and it is also as an attractive strategy toward the total synthesis of complex natural products^{28,29,30}.”

The 1,3-dipolar cycloaddition of transient carbonyl ylides generated from carbene has been summarized in reviews and it has never been used in type II cycloadditions. Two new references have been added to ref. 29 and 30.

[29]. Padwa, P. *Intramolecular cycloaddition of carbonyl ylides as a strategy for natural product synthesis. Tetrahedron* **67**, 8057-8072 (2011).

[30]. Davies, H. M. L. & Alford, J. S. *Reactions of metallocarbenes derived from N-sulfonyl-1,2,3-triazoles. Chem. Soc. Rev.*, **43**, 5151-5162 (2014).

Question 2: The cycloaddition described herein is different from the previous examples because it does not produce an anti-Bredt double bond. This is the reason why smaller rings can be accessed compared to the other type II cycloadditions and this should be stated.

Answer:

We highly appreciate this reviewer's comments. We have stated this reason in the text with highlighting in Page 4, as the following:

"The anticipated cycloaddition described herein is different from the previous examples because it does not produce anti-Bredt double bond. This is the reason why smaller rings could be accessed compared to the other type II cycloadditions."

Question 3: There should be a mention of the background about the uses of carbenes derived from N-sulfonyltriazoles in cycloaddition reactions.

Answer:

We highly appreciate this reviewer's comments. The background about the uses of carbenes derived from N-sulfonyltriazoles in cycloaddition reactions have been added in Page 5 of the text with highlighting, as the following:

"The carbenes derived from the corresponding N-sulfonyltriazole have been used in cycloaddition reactions to synthesize fused all-carbon ring systems."

The references have been added to refs. 35-37 accordingly.

[35] Spangler, J. E. & Davies, H. M. L. *Catalytic asymmetric synthesis of pyrroloindolines via a Rhodium (II)-catalyzed annulation of indoles. J. Am. Chem. Soc.* **135**, 6802-6805(2013). [36] Li, Y., Zhang, Q. Y., Du, Q. C., Zhai, H. B. *Rh-catalyzed [3+2] cycloaddition of 1-sulfonyl-1,2,3-triazoles: access to the framework of aspidosperma and kopsia indole alkaloids. Org. Lett.* **18**, 4076-4079 (2016); [37] Yuan, H., Gong, J. X., Yang, Z. *Stereoselective synthesis of oxabicyclo[2.2.1]heptenes via a tandem dirhodium(II)-catalyzed triazole denitrogenation and [3 + 2] cycloaddition. Org. Lett.* **18**, 5500-5503(2016).

Question 4: There is a big analysis of all the possible carbene derived products that could

have been formed. However, most of these products were not especially likely so the discussion is a bit of an exaggeration. The formation of the carbonyl ylide was highly likely and if the alkene was suitably placed the cycloaddition would be expected to occur. Li's recognition of the opportunity was clever but the outcome is what one would have expected.

Answer:

We highly appreciate this reviewer's comments. After careful consideration, we have revised the discussion of the possible carbene derived products that could have been formed in Page 5, and the impossible "aldehyde cyclization product with an anti-Bredt double bond" has been removed in Fig.1C in the Text.

Additional question: Finally, the draft in its current form has a lot of grammatical errors and clumsy sentences. The schemes look great but someone needs to go through the text very carefully.

Answer: We highly appreciate this reviewer's comments. We have carefully revised the grammatical errors and inappropriate statement in the text.

Reviewers 3:

Recommendation: I believe it can be published as is.

Comments: The manuscript authored by Houk, and Li et al. describes a new synthetic approach to bridged polycyclic compounds based on the application of catalytic intramolecular [3+2] cycloadditions. A wide range of functionalized [m.n.2] bridged scaffolds can be accessed using this method. The utility of this practical methodology was fully demonstrated by completion of the 17-step asymmetric total synthesis of nakafuran-8 highlighted by the rapid assembly of the challenge all carbon [4.2.2] bridged ring system. Besides, detailed mechanistic study of the regioselectivity, chemoselectivity, and diastereoselectivity of the [3+2] cycloaddition was provided through DFT calculations. Since synthesis of bridged medium-sized bicyclo[m.n.2] rings via Type II intramolecular [3+2] cycloadditions has been even rarer, this work is of great importance in the field of cycloaddition reactions and therefore the reviewer would be happy to support its publication in Nature Communications. This work shows a complete story from method development, mechanistic study and synthetic applications. I believe it can be published as is.

Answer: We highly appreciate this reviewer's comments.

Reviewers' Comments:

Reviewer #1:

Remarks to the Author:

The authors have properly addressed my concerns and the manuscript now deserves publication in its current form.

Reviewer #2:

Remarks to the Author:

The changes made in the revised document suitably address the original concerns